# Effects of Active Breaks on Physical Literacy: A Cross-Sectional Pilot Study in a Region of Spain

**DOI:** 10.3390/ijerph19137597

**Published:** 2022-06-21

**Authors:** María Mendoza-Muñoz, Violeta Calle-Guisado, Raquel Pastor-Cisneros, Sabina Barrios-Fernandez, Jorge Rojo-Ramos, Alejandro Vega-Muñoz, Nicolás Contreras-Barraza, Jorge Carlos-Vivas

**Affiliations:** 1Research Group on Physical and Health Literacy and Health-Related Quality of Life (PHYQOL), Faculty of Sport Sciences, University of Extremadura, 10003 Caceres, Spain; 2Departamento de Desporto e Saúde, Escola de Saúde e Desenvolvimento Humano, Universidade de Évora, 7004-516 Evora, Portugal; 3Research Group on Physical and Health Literacy and Health-Related Quality of Life (PHYQOL), Faculty of Medicine and Health Sciences, University of Extremadura, 06800 Badajoz, Spain; 4Promoting a Healthy Society Research Group (PHeSO), Faculty of Sport Sciences, University of Extremadura, 10003 Caceres, Spain; raquelpc@unex.es (R.P.-C.); jorgecv@unex.es (J.C.-V.); 5Social Impact and Innovation in Health (InHEALTH) Research Group, Faculty of Sport Sciences, University of Extremadura, 10003 Caceres, Spain; sabinabarrios@unex.es (S.B.-F.); jorgerr@unex.es (J.R.-R.); 6Public Policy Observatory, Universidad Autónoma de Chile, Santiago 7500912, Chile; alejandro.vega@uautonoma.cl; 7Facultad de Economía y Negocios, Universidad Andres Bello, Viña del Mar 2531015, Chile; nicolas.contreras@unab.cl

**Keywords:** active breaks, physical literacy, playful games, schoolchildren

## Abstract

Several studies have shown that active breaks (AB) lead to improvements in physical fitness, daily steps taken and participants’ health. However, there are no studies that have evaluated how they affect physical literacy (PL). Aims: Therefore, this study examined the effects of a 4-week recreational AB program based on games whose main objective was to improve motivation and motor skills’ improvement in PL in schoolchildren. Method: A quasi-experimental pilot study was conducted with PL assessments before and after a 4-week recreational AB program. Results: Fifty-seven schoolchildren (10.28 ± 0.43 years) participated in the study, 29 in the control group and 28 in the experimental group. An improvement in PL was found between the experimental and control groups after the intervention (*p* = 0.017). Moreover, the experimental group also improved (*p* < 0.001) PL after the intervention. More specifically, within the domain of PL, improvements were found after the intervention in the experimental group in the domains of physical competence (*p* < 0.001), motivation and confidence (*p* < 0.001) and knowledge (*p* < 0.001) but not in the domain of daily activity (*p* = 0.051). Conclusion: The application of an AB program based on playful games, during four weeks, produced an increase in scores in the general PL level, as well as in the domains of physical competence, motivation, and knowledge and understanding in schoolchildren.

## 1. Introduction

Sedentary lifestyles present one of the most important challenges in current society, being considered the disease of the 21st century [1,2]. Many studies [3,4,5] have analysed physical inactivity behaviours in children and young people, reporting worrying results. Focusing on Spain, the PASOS study [3] showed that only 36.7% of children and young people met the WHO recommendations for daily physical activity [6]. This study also reported that females present higher percentages of inactivity compared to their counterpart males (70.1% vs. 56.1%); in addition, adolescents showed greater inactivity than children (69.9% vs. 56.1%). The ALADINO 2015 report [5] revealed similar results, indicating that around 68% of children aged between 6 and 9 years, performed less than 4 h of physical activity (PA) per week. Subsequently, the 2019 ALADINO report update ratified its previous report data, stating that the percentage of boys with less than 4 h per week of PA was 69.7%, while it was 80.6% for girls [4]. Thus, all previous studies that have analysed physical inactivity in children and young people agree with the high degree of sedentary lifestyles in our society and, more specifically, in children and adolescents; this being especially worrying in the case of girls.

Hence, sedentary behaviour is one of the challenges in current society due to the numerous problems that physical inactivity can lead to and the risk it poses both in the short- and long-term for health. The health benefits of PA and its importance in disease prevention have already been demonstrated [7]. In this way, environmental factors and lifestyle will need to play a key role in the adherence to PA and, consequently, in health prevention, with physical exercise becoming an important factor [8].

Focusing on the factors that can influence an improvement in adherence to PA, we should highlight the role that physical literacy (PL) can play in it, improving physical fitness and its consequent benefits. PL was defined, in the Bulletin of the International Council of Sport Science and Physical Education of the United Nations Educational, Scientific and Cultural Organization, as the motivation, confidence, physical competence, knowledge and understanding to value and participate in a physically active lifestyle [9]. Just as reading, writing, listening and speaking combine to formulate linguistic literacy, PL is a progressive journey in which the different components interact holistically to facilitate a life of participation in and enjoyment of PA. Thus, the positioning of PL as a determinant of health is not a novel idea, but little attention has been paid to what it means and its implications. Although few studies have investigated the association between PL and health, it has already been shown to be related to body composition [10], physical fitness, blood pressure and health-related quality of life [11].

Physical literacy can be addressed both within and outside the educational setting. Castelli et al. [12] highlight that within the educational setting, curricula can contribute to PL in different ways, differentiating between structured activities, unstructured or informal physical activities (recess), or content-rich physical activity instruction (combining academic concepts with movement). In this regard, a number of studies are beginning to address PL both within the PE classroom [13,14] and during out-of-school periods [15,16].

In this context, it is important to refer to active breaks as a strategy to prevent the previously mentioned challenges. Briefly, active breaks consist of short bursts of physical activity performed in the school as a break from learning tasks [17], the main aim of which is to activate students cognitively, with special emphasis on improving executive functions. Thus, active breaks may provide an attractive strategy for teachers to incrementally increase children’s daily physical activity during school hours [18,19] while simultaneously improving classroom behaviour outcomes. Previous studies have reported that active breaks (AB) led to improvements in physical fitness [20,21], social and cognitive interaction [22], the number of daily steps carried out [23], slight changes in the BMI [20] and even improvements in participants’ health [21]. Thus, such practices become even more important and necessary in a school environment where students spend around 6 h physically inactive sitting at a desk; only having a 30-min break and the 5-min times between classes as free time to have active behaviours, in addition to the two hours per week of physical education classes. However, there are no studies that have evaluated how it affects PL. Additionally, some studies observed that PL can be improved within physical education classes [13,14] or in the extracurricular field [15,16]. Thus, an intervention during rest periods could lead to an improvement in PL.

There are no studies in Spain that have implemented programs directed at improving PL. However, we consider this multidimensional evaluation and intervention model to be of great interest, and, therefore, this project aims to be a starting point to stimulate empirical research on PL since the development and improvement of its levels and every one of its domains will allow children to participate in a structured and full PA, adopting active lifestyles from an early age until adulthood, with the health benefits that this entails.

Therefore, this study examined the effects of a 4-week recreational AB program based on games whose main objective was to improve motivation and motor skills in PL in schoolchildren.

## 2. Materials and Methods

### 2.1. Design

A quasi-experimental pilot study was conducted with pre- and post-intervention assessments to assess the effect of a 4-week recreational AB program.

### 2.2. Participants

A total of 57 schoolchildren from a school in the province of Badajoz (Extremadura, Spain), aged between 9 and 12 years, participated, of whom 29 were girls (50.87%) and 28 boys (49.13%) with a mean age of 10.28 ± 0.43 years.

Participants met the following eligibility criteria: (1) aged 8–12 years; (2) authorised by their parents or legal guardians; (3) children agreed to participate in the study and (4) not having pathologies that prevent participation in physical fitness tests or practice.

### 2.3. Ethics Approval

This study was approved by the Bioethics and Biosafety Committee of the University of Extremadura (approval number: 23/2021). We followed the updates of the Declaration of Helsinki, modified by the 64th General Assembly of the World Medical Association (Fortaleza, Brazil, 2013) and the Law 14/2007 on Biomedical Research.

### 2.4. Procedures

A presentation of the study was made to the staff responsible for the school and the teacher responsible for the physical education area, who gave their approval. After this, two classes were selected from the school, one of which was assigned as the experimental group (EG) and the other as the control group (CG). Both the assessments and the intervention were conducted by qualified staff who were part of the study, the school staff did not participate in the study. Children in the control group only participated in the initial and final assessments. Participants in the control group did free activity as in any regular recess. The children in the experimental group participated in the assessments and also took part in an AB program for one month. The AB took place on school break days and lasted about 15 min. During these sessions, PA was carried out through playful games. The AB protocol was adapted based on previous research that developed AB programs based on alternative, popular and traditional games [24], free games [25] or pre-sports games [26]. The games had mostly a cooperative and a competitive component, due to the motivational benefits these can bring [27]. The sessions had the following distribution (Table 1):

Catch the flag: the participants were divided into two teams and the indoor football field was divided into two zones. A “flag” was created with a cone, a pike and a flagpole. Each team placed its flag inside the semi-circular areas of the indoor football field. The objective was to steal the opposing team’s “flag” without being touched by their opponents. Each team, in turn, could not enter the area where their flag was located. Once the participants entered the semi-circular area where the opposing flag was located, they could not be touched and had to leave with it and move it to the end line of their field.

-Level 2: the difficulty was extended to include two flags so that no point was scored until both flags were stolen at the same time, and it could not be the same person who stole both flags.-Level 3: three flags per team were included. In this case, no two people from the same team could remain inside the opposing area, so until one of them stole a flag, another teammate could not enter to steal the next one.

Rock, paper, scissors [28]: The participants were divided into two teams, and a circuit with hoops was set up on the floor. The participants had to jump through the hoops to reach the end of the circuit. Along the circuit, they met their teammates from the other team with whom they had to play a game of “rock, paper, scissors”. The winner could continue to move along the circuit and the loser had to return to the tail of his team until he could start again. Each participant who reached the end of the circuit scored one point.

-Level 2: only jumping movements with two feet together were allowed inside the hoops. In this way, they progress more slowly and need more coordination and balance.-Level 3: only jumping with one foot together was allowed.

Dodge ball [29]: the participants were divided into two groups and the space was divided into 4 zones (two central fields separated by a line and two lateral cemeteries, behind the central fields); the material used was a foam rubber ball. The game consisted of throwing the ball from the central field of one team to the cemetery of the same team to hit a teammate of the opposing team who was located in the central field of the opposing team. When this happened, this teammate was “eliminated” and moved from the central field to the cemetery area of his team. This continued until one team ran out of players in the centre field [30].

-Level 2: the level of difficulty was increased by introducing two balls during the game.-Level 3: based on the premise of level two, a premise was added that if an opposing team’s throw was intercepted without the ball falling to the ground, a teammate could be saved and returned to the centre of the field.

Fox hospital: The participants were divided into two groups. Each group wore dungarees of a “green” or “red” colour. Each participant was a “fox”, who had to have dungarees or handkerchief hanging on his trousers as a tail. For this game, at the signal “go”, the foxes, without leaving the sports area of the indoor football pitch, had to prevent their teammates from the opposing team from removing their tails. If the foxes had their tails taken away, they had to take refuge inside the goal (fox hospital) until a fellow fox from their team could bring them another tail to put back on and continue playing. For each extra tail they obtained, a point was added at the end of the game time.

-Level 2: the number of queues was reduced.-Level 3: the playing space was reduced.

Cards game: participants were divided into 4 teams, which were placed in rows. Each participant of each team had to run individually to a table located about 30 m away, and once there they found a series of cards (10) and had to place the card they were holding on to their corresponding (partner) card and return to the end of the line for the next partner to come out. The first team to complete all pairs won (all cards were images related to PA or healthy lifestyle habits).

-Level 2: the number of cards was increased from 10 to 15.-Level 3: the time they could look at the cards on the table was limited to 5 s, if they failed, they had to return with their card to the start to try again.

Figure 1 displays a schematic representation of an active break sequence.

### 2.5. Measures

Anthropometric measurements were carried out under standardised conditions, following WHO recommendations for the development of the Childhood Obesity Surveillance Initiative (COSI) [31] and the ALADINO report [32]. Before the measurements, participants removed their shoes and socks, as well as any heavy clothing (coats, jumpers, jackets, etc.). In addition, they emptied their pockets, removed their belts and any other accessories they were wearing (headbands, pendants, etc.).

Height was measured with a height measuring device (Tanita Tantois, Tanita Corporation, Tokyo, Japan). It was placed on a vertical surface so that the measuring scale was perpendicular to the ground. The participants stood with shoulders balanced and arms relaxed along the body. Measurements were taken in cm to the nearest mm.

Bodyweight was measured with a bioimpedance meter (Tanita MC-780 MA, Tanita Corporation, Tokyo, Japan). The assessment was performed in “standard mode” by entering the participant’s age, sex and height. Bodyweight was recorded in kg, with an accuracy of 100 g.

From these two measurements, BMI was determined using the equation BMI = bodyweight/height^2^.

The level of PL was assessed using the Canadian Assessment of PL Development (CAPL-2) [33]. This assessment included 4 domains: (1) daily physical activity behaviour, (2) physical competence, (3) motivation and confidence, and (4) knowledge and understanding. Each domain was scored and consisted of different tests.

Each of the domains that make up the CAPL-2 and how they are scored are described below:

Daily Behaviour. The total score was made up of scores from two components: step counts obtained using an activity wristband (Xiaomi mi Band 3, Xiaomi Corporation, Beijing, China), which recorded steps for a full week, and a self-reported question on the number of days active for at least 60 min. The total score for this domain was therefore composed of the score obtained from the number of steps recorded and the score assigned to the response to the self-reported question on the number of minutes of weekly activity.

Physical competence. The final score for this domain was reached with the sum of the scores of three components:-Plank [34], consisted of holding the plank position for as long as possible.-Progressive Aerobic Cardiovascular Endurance Run (PACER) [35], which made it possible to determine cardiorespiratory competence using a 20-m running test (out and back), following an acoustic signal that determined the intensity of the test.-Canadian Agility and Movement Skill Assessment (CAMSA) [36], allowed the participants to test their motor skills through an agility circuit, which included throwing, jumping and moving actions.

All these tests were evaluated with a possible score from 1 to 10 points, giving a total score of 30 points for this domain.

Motivation and confidence. This was assessed using the CAPL-2 motivation and confidence questionnaire [33], which assessed participants’ confidence in the ability to be physically active, and motivation to participate in PA. The score was obtained by summing four different dimensions assessed from 1 to 7.5 points: intrinsic motivation, competition, predilection and appropriateness. The total domain score ranged from 1 to 30 points.

Knowledge and understanding. This domain assessed knowledge concerning physical activity [37]. Its score was obtained from a questionnaire that included five questions, four of which were multiple-choice questions and were scored from 0 to 1. The last question consisted of fill-in-the-blanks questions to complete a story and was scored from 1 to 6.

Finally, the CAPL-2 numerical scores ranged from 0 to 100 points.

### 2.6. Statistical Analysis

Data analysis was conducted using IBM Social Sciences software (SPSS, version 25.0, Armong, IL, USA). Descriptive statistics are presented as mean and standard deviation (SD) for all socio-demographic and dependent variables. Normality and homogeneity of data were examined through Levene’s and Kolmogorov–Smirnov tests, respectively. Then, a group × time analysis of variance (ANOVA) was conducted to analyse within-group and between-group differences in all dependent variables. Alpha level was set at *p* ≤ 0.05.

## 3. Results

Table 2 shows the socio-demographic characteristics of the participants in both the control and experimental groups. No significant differences were found for any of the socio-demographic variables between the control group and the experimental group (*p* = 0.195 to *p* = 0.789).

Table 3 shows the CAPL-2 scores for both the control and experimental groups, as well as the intergroup differences. In addition, pre- and post-intervention scores for each group and the differences between the two assessments can be seen. The results reported no differences between the control and experimental groups in the pre-test, except the motivation domain, in which significant differences were detected in the competence subdomain (*p* = 0.016). In the case of the post-test, significant improvements were detected in the experimental group concerning the control, except the domain of daily activity (*p* = 0.741).

Concerning total PL, a significant improvement was found between the experimental and control groups after the intervention (*p* = 0.017). Furthermore, within-group, the experimental group also reported a significant improvement between the pre and post assessment (*p* < 0.001), which was not the case in the control group (*p* = 0.344).

More specifically, in the domain of daily activity, no significant differences were reported in the experimental group (*p* = 0.051), although significant differences were reported in the control group (*p* = 0.005); however, these differences were caused by a worsening of the score in this domain (from 16.50 to 15.32 points).

Regarding the domain of physical competence, significant improvements were found in the experimental group for the total domain (*p* < 0.001), as well as for its component tests (<0.001), except for the plank test (*p* = 0.260). However, for the control group, no significant differences were found.

For the motivation and confidence domain, all subdomain scores were significantly improved for the experimental group (*p* < 0.001 to *p* = 0.023), in contrast to the control group, which remained at similar scores or even significantly worsened, such as the subdomain of “adequacy” (from 5.28 to 5.10 points).

Finally, in the knowledge domain, there was a significant improvement in the experimental group (*p* < 0.001), which was also not reflected in the control group (*p* = 0.673).

## 4. Discussion

Currently, there is a great deal of interest in the concept of physical literacy due to its comprehensive nature on the development of physical activity. In this regard, several studies have reported data on the improvement of physical literacy following a physical activity program both within physical education classes [13,14] and in the out-of-school environment [15,16]. However, no studies have been found in which physical literacy has been explored after a program of AB.

The present study examined the effects of an AB program on the level of physical literacy in children aged 8–12 years. The main findings show an increase in physical literacy scores in the group that underwent the AB program, while in the control group they remained unchanged and even worsened in some domains.

Once the program was implemented, the physical literacy of the experimental group increased over the control group, except for the daily activity domain. Few studies have explored the evolution of physical literacy after a physical activity program, but in this line, Coney et al. [13] did not see either an improvement in this domain or the motivation domain after a 12-week physical activity program.

However, we have found studies that have explored the improvements that an AB program could bring about in any of the domains that make up physical literacy. More specifically, if we explore the evolution of each domain in both groups, after the AB program, we could observe that concerning the daily activity domain, our study did not report differences in the experimental group after the program, although they were negative in the control group, i.e., there was a decrease in daily steps in this group between the initial and final evaluation, and therefore a worsening in the score on this test. This is in line with previous studies evaluating physical literacy pre and post physical activity-based interventions, which also failed to detect improvements in this domain. [13]. However, Masini et al. [38] found an increase in the number of steps after a 14-week program of AB, this success may be due to the increased stimulus, as the breaks were performed twice a day (morning and afternoon). Therefore, an increase in both the time and the number of daily stimuli could favour the improvement of this domain. However, although the results of this study do not report significant improvements in the number of steps, a significant improvement can be seen in the report on the number of days per week practicing physical activity.

Regarding the domain of physical competence, significant improvements were found in the experimental group for the total domain, as well as for the tests that compose it, except for the plank test; in this sense, [39] developed an 8-week program to investigate the effects of AB on physical fitness performance using specific muscle strengthening exercises and also found no improvement. This may be because two to three weekly strength training stimuli would be necessary for muscle improvement [40]. Therefore, in our study, the cardiorespiratory component, speed and agility prevailed in the activities carried out. It would be necessary to introduce, in future studies, activities where muscle strength is developed in compliance with the guidelines that lead to an improvement.

For the motivation and confidence domains, all subdomain scores were significantly improved, in contrast to the control group, which remained at similar scores or even significantly worsened, such as the subdomain of “adequacy”.

In this sense, other studies that have carried out interventions based on physical activity have reported improvements at the motivational level, both related to intrinsic motivation, predilection, adequacy, and competence [41]. In this line, and concerning the daily activity domain, Franco et al. [42] suggested that interventions that, in a physical education context, improve students’ perception of competence may favour their intention to engage in physical activity; in addition, Van der Horst et al. 2007 [43] reported that cognitive attitude, affective attitude, self-efficacy and perceived competence could be positive factors for increasing participation in PA outside of the school context. Therefore, more specific work on the motivational domain of PC in future AB programs could increase the daily activity domain, as a direct relationship between the two domains has also been found [44]. In short, generating interventions in different content that improve students’ predisposition towards physical education could contribute to them adopting a more active lifestyle [41].

In addition to the abovementioned benefits of school-based physical activity interventions, other studies have reported positive effects on children’s self-reported psychological well-being, social and peer support, and the school environment [45]. Thus school could be an ideal setting for increasing physical activity levels, as it is the place where children spend most of their day during childhood [46]. Thus, this provides an opportunity for teachers (both generalist and PE teachers) to help anchor experiences in a didactic pedagogical and curricular framework in which pupils’ learning and health promotion processes are developed [46].

However, the motivations to be more physically active change as children grow older, evolving to adapt to their evolving needs, such as fitness, weight control, social victimisation, health benefits or enjoyment [47,48,49]. Therefore, one of the main applications of physical literacy assessment can be the detection of deficits in the different domains that make up physical literacy, to target appropriate interventions.

More specifically, implementing programs, such as AB, that can improve physical literacy by assessing and monitoring children’s physical literacy from primary to secondary school could provide an insight into the evolution of physical literacy at different stages, and thus allow for interventions to be tailored to children’s and/or adolescents’ needs, for a physically literate child can move skilfully and confidently in a wide variety of physically challenging situations, can read the physical environment, anticipate possible movement needs and respond intelligently and imaginatively [9] to difficulties encountered. In contrast, a child who has not yet developed a high level of physical literacy will try to avoid physical activity whenever possible, will have minimal confidence in their physical ability and will not be motivated to participate in structured physical activity [50]. The assessment and development of physical literacy, therefore, could help to explain why children do or do not participate in physical activity [51], trying to understand how physical activity influences them and may help them to lead a more active life [44].

The main added value of this study is the acquisition of new knowledge concerning physical literacy from an immediate perspective of practical and direct application, which represents a scientific advance in terms of improvement and adherence to health-related PA, as well as the prevention of diseases associated with inactivity such as overweight and obesity. On the other hand, the search for a solution to the social problems regarding PA, such as the difficulty of accessing PA programs specially related to health, above all for socio-economic reasons, can be solved thanks to this proposal that facilitates access to an accessible method through the school day. Thus, given that it is a low-cost program that can be easily standardised by levels of difficulty (adapting to the different levels of PA), its transfer and implementation would not be a problem either in the public sphere, with the possibility of including the program in the educational sphere within public education programs, or in the private sphere, where the potential for its transfer and implementation in training centres is identified, as well as possibly in workplaces, especially in those where work is mostly sedentary.

Specifically, this study has been developed at school age. However, data have recently been published on the validity of the CAPL assessment in adolescents aged between 12 and 16 years [52]. Thus, one of the main lines of future research is the assessment of physical literacy and the development of a program of AB to improve it at adolescent age, explore the differences between stages, and be able to adjust interventions as early as possible, thus trying to avoid the abandonment of sports practice and the health consequences that this could entail, as it has been shown in numerous studies that the general level of physical activity is lower among adolescents compared to children [3,53]. Another line for the future that could be addressed is the effect of this program on school learning in other areas, i.e., to see if there is an improvement in other areas such as language or mathematics when this type of activity is carried out during breaks, since recent reviews of the literature on programs similar to the one in this study reports that including PA in the school timetable through AB could positively affect attention or academic achievement [54,55].

The findings are strengthened by the use of an objective assessment of physical literacy, validated for the assessment of physical literacy in the target population (8–12 years) [33]. However, it should be noted that this study had some limitations, among which is the limitation of the sample, which was small and non-randomised. The non-randomisation of the sample was due to measures imposed by the school administration due to COVID-19 pandemic restrictions, which did not allow students from different classrooms to share the same space during recess. Previous studies that developed similar programs in schools have already reported this difficulty [38,56].

## 5. Conclusions

The application of an AB program based on playful games and aimed at motivation and motor skills work, for four weeks, produced an increase in scores in general physical literacy levels as well as in the domains of physical competence, motivation, and knowledge and understanding in schoolchildren aged 8–12 years. Moreover, these scores were higher than those of children who did not take part in any additional program and continued with their usual school routine. Thus, the implementation of an AB program such as the one proposed in this study could serve to prevent and reduce physical inactivity and sedentary time at school and thus combat the problems that these behaviours can lead to. However, considering that this study is one of the first to evaluate the evolution of physical literacy after implementing an AB program, future studies are needed in this area, where interventions can be more specifically oriented towards the development of the domains that make up physical literacy and where interventions of longer duration can be carried out.

An adequate way to introduce and implement active breaks at school would be to carry out activities during the 5 min break between classes or by starting each class with a brief activation of the students in the classroom by performing activities to remind them of the contents taught in the previous class of that subject while they carry out motor activities. Another possibility would be to take advantage of the spaces in the school outside the classroom to design a route for students to go out for aerobic physical activity and practice basic motor skills for 8–10 min after long periods of class. Ideally, however, all teachers in the school should be trained to acquire basic knowledge about the concept and applications of active breaks so that they can easily implement it.

## Figures and Tables

**Figure 1 ijerph-19-07597-f001:**
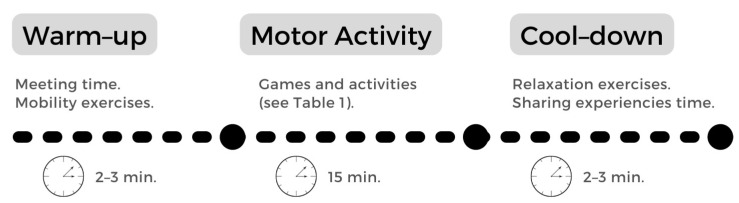
Schematic representation of a session day.

**Table 1 ijerph-19-07597-t001:** Distribution of the contents of the sessions per day during the 4 weeks of intervention.

		Monday	Tuesday	Wednesday	Thursday	Friday
Week 1	Level 1	Catch the flag	Rock, paper, scissors	Dodge ball	Fox hospital	Cards game
Week 2	Level 1	Catch the flag	Rock, paper, scissors	Dodge ball	Fox hospital	Cards game
Week 3	Level 2	Catch the flag	Rock, paper, scissors	Dodge ball	Fox hospital	Cards game
Week 4	Level 3	Catch the flag	Rock, paper, scissors	Dodge ball	Fox hospital	Cards game

**Table 2 ijerph-19-07597-t002:** Total sample characteristics and stratified by group and gender.

	All Participants	Experimental Group (*n* = 29)		Control Group (*n* = 28)		Between-Group Comparison
Gender		Male	Female	Male	Female	
*N* (%)	57 (100)	15 (51.7)	14 (48.3)	13 (46.4)	15 (53.6)	
	Mean (SD)	Mean (SD)		Mean (SD)		*p*
Age (years)	10.25 (0.43)	10.17 (0.38)		10.31 (0.48)		0.195
Height (cm)	143.62 (6.95)	141.96 (7.34)		145.33 (6.20)		0.670
Weight (kg)	40.43 (10.73)	38.77 (8.96)		42.15 (12.24)		0.334
BMI (kg/m^2^)	19.14 (3.80)	19.09 (3.05)		19.20 (4.49)		0.780
Fat Mass (%)	23.67 (6.92)	26.81 (5.99)		25.88 (7.97)		0.607

**Table 3 ijerph-19-07597-t003:** Outcome of physical literacy measures at baseline, follow-up and changes at 4 weeks.

	Experimental Group (*n* = 29)							Control Group (*n* = 28)								
	Baseline		Follow-Up		Change		Within-Group	Baseline		Follow-Up		Change		Within-Group	Between Group Pre	Between Group Post
	Mean	SD	Mean	SD	Mean	SD	*p*	Mean	SD	Mean	SD	Mean	SD	*p*	*p*	*p*
DB Domain (points)	17.17	7.39	17.96	7.06	0.79	−0.33	0.051	18.96	6.89	17.75	6.16	−1.21	−0.73	0.005	0.473	0.741
Self-reported question (points)	3.17	1.46	3.31	1.46	0.14	0	0.044	2.46	1.64	2.42	1.54	−0.04	−0.1	0.308	0.094	0.026
Diary steps (points)	14.00	7.37	14.65	6.96	0.65	−0.41	0.108	16.50	6.22	15.32	5.49	−1.18	−0.71	0.008	0.263	0.854
PC Domain (points)	14.24	5.59	18.22	5.33	3.98	−0.26	<0.001	14.84	5.43	14.53	5.05	−0.31	−0.38	0.635	0.687	0.010
CAMSA (points)	5.45	1.85	6.94	1.52	1.49	−0.33	<0.001	5.6	1.42	5.93	1.59	0.33	0.17	0.234	0.714	0.012
Plank (points)	5.44	3.1	4.88	2.14	−0.96	−0.96	0.260	5.39	3.22	3.75	2.15	−1.64	−1.07	0.375	0.948	0.011
PACER (points)	3.34	2.00	6.79	2.73	3.45	0.73	<0.001	3.78	2.1	4.8	2.85	1.02	0.75	0.864	0.426	0.204
M and C Domain (points)	23.97	1.79	25.11	1.31	1.14	−0.48	<0.001	22.27	3.08	21.75	2.69	−0.52	−0.39	0.011	0.016	<0.001
Predilection (points)	5.64	0.75	5.93	0.63	0.29	−0.12	0.003	5.5	1.13	5.34	1.16	−0.16	0.03	0.102	0.603	0.022
Adequacy (points)	5.41	1.04	5.63	0.8	0.22	−0.24	0.023	5.28	1.10	5.10	.93	−0.18	−0.17	0.037	0.659	0.028
Intrinsic motivation (points)	6.70	0.67	7.01	0.49	0.31	−0.18	0.001	6.26	1.09	6.10	1.0	−0.16	−0.09	0.081	0.086	<0.001
Competence (points)	6.20	0.90	6.62	0.64	0.42	−0.26	0.001	5.21	1.59	5.16	1.11	−0.04	−0.48	0.653	0.006	<0.001
K and U domain (points)	7.00	1.30	7.72	1.19	0.72	−0.11	<0.001	6.67	1.41	6.6	1.28	−0.07	−0.13	0.673	0.385	0.002
Overall Physical Literacy (points)	61.19	11.96	68.30	10.85	7.11	−1.11	<0.001	61.68	14.68	60.72	11.90	−0.96	−2.78	0.344	0.893	0.017

DB: daily behaviour domain; PC: physical competence domain; CAMSA: Canadian Agility and Movement Skill Assessment; PACER: Progressive Aerobic Cardiovascular Endurance Run; M and C: motivation and confidence domain; K and U: knowledge and understanding domain.

## Data Availability

The datasets used during the current study are available from the corresponding author on reasonable request.

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
