# Peer review of "Effects of Active Breaks on Physical Literacy: A Cross-Sectional Pilot Study in a Region of Spain"

_ijerph, 2022, doi:10.3390/ijerph19137597_

Round 1

Reviewer 1 Report

The work is highly appreciated. In addition, the paper mentioned an important topic of children and shown the difficulty of keep the attention during hour-long class. For this reason, it has a valuable and socially useful content. The manuscript is very well written and structured. I have some suggestions for the authors.

-       For future investigations should be indicated at the end of the abstract as well as in the conclusions section some directions about the results in relation to children.

Introduction:

-  The work covers the gap of AB in the literature. The most novelty topic of the manuscript is the psychomotor vigilance task and active breaks in students of Primary School. However, the absence of physical education context in the Introduction is evident. Maybe, we would suggest inserting a different paragraph about this fact.

- The most recent updates should be mentioned in the Introduction, ¿do we have new articles in the last two years? check:

- The introduction doesn’t explain the basic concepts used in the paper, such as the importance of this topic. Some of these are presented in the methodology section, in terms of how they were measured, however it would have been necessary to clarify their general content in the introduction section

Methods.

-       In reference to organization of the AB, Could AB applied from the point of view of the organization of an educational center? And the AB, could be applied in another educational environment?

-        Besides to Physical education teachers, would it be necessary for classroom teachers to be trained in the implementation and start-up of active breaks? Can you elucidate in the manuscript.

-       To better understand the intervention the authors should be describe the Active breaks – Very important here is the moderators that have influence in the different conditions. Which moderators were taken into account?

-       Considering the subject of the paper, the introduction should have mentioned the main cultural characteristics of scholar environmental.

-       I suggest a figure with a schematic representation of a test day to understand correctly the work and procedures.

Results.

-       Results: I was elaborated well. In this case, the manuscript providing immediate results inside the classroom with cognitive task, something different from what is found in the literature that normally use questionnaires. The results of ANOVA for RT and RPE were quite clarifying and robust.

-       Elucidate the table, add the confidence intervals 95% upper and lower and add note under the table to clarify the information.

Discussion

-       Discussions is appropriate, making reference to the results of other studies, but the theoretical and practical implications of the research are vaguely mentioned. It necessary improve it.

References

-       The references used in the present manuscript were sectioned correctly and help to further clarify the interest of the present research topic. Update and check the last two years

Other

-       Last, how would you start to be implementing the AB in classroom? Where should you start?

-       Be consistent with the terms. i.e.: Active-breaks?

Author Response

Dear Reviewer,

Thank you for your review of our manuscript. We have carefully considered your comments and believe that the quality of the paper has improved after incorporating your suggestions. Below are our responses to your suggestions:

The work is highly appreciated. In addition, the paper mentioned an important topic of children and shown the difficulty of keep the attention during hour-long class. For this reason, it has a valuable and socially useful content. The manuscript is very well written and structured. I have some suggestions for the authors.

-       For future investigations should be indicated at the end of the abstract as well as in the conclusions section some directions about the results in relation to children.

Introduction:

-  The work covers the gap of AB in the literature. The most novelty topic of the manuscript is the psychomotor vigilance task and active breaks in students of Primary School. However, the absence of physical education context in the Introduction is evident. Maybe, we would suggest inserting a different paragraph about this fact.

                - Authors’ response: Thank you for your comment, we have added the information as you suggested.

- The most recent updates should be mentioned in the Introduction, ¿do we have new articles in the last two years? check:

                - Authors’ response: Thank you for your comment, we have added new references.

- The introduction doesn’t explain the basic concepts used in the paper, such as the importance of this topic. Some of these are presented in the methodology section, in terms of how they were measured, however it would have been necessary to clarify their general content in the introduction section

                - Authors’ response: Following your suggestion, we have include a brief explanation about the basic point of physical literacy into de introduction. If you consider a more specific explanation is needed, let know us please.

 Methods.

-       In reference to organization of the AB, Could AB applied from the point of view of the organization of an educational center? And the AB, could be applied in another educational environment?

                - Authors’ response: The study was carried out in educational centres, during the long break between classes, so we consider that it is possible that it could be replicated in other educational centres, as there is also a multitude of literature that shows successes in similar programmes. Furthermore, we believe that it could be developed in other types of centres due to the limited resources and space needed for its development, as well as the possibility of adapting the activities to different ages and groups. Based on your comments, we have added a paragraph in the discussion section in which we refer to this.

-        Besides to Physical education teachers, would it be necessary for classroom teachers to be trained in the implementation and start-up of active breaks? Can you elucidate in the manuscript.

- Authors’ response: We appreciate you comment. Thus, we have highlighted the needed to train and raise awareness all the teachers in the concept and applications of active breaks at the end of the conclusions section.

-       To better understand the intervention the authors should be describe the Active breaks – Very important here is the moderators that have influence in the different conditions. Which moderators were taken into account?

- Authors’ response: Following your suggestion we have defined the active breaks and highlight their benefit related to the topic of this manuscript.

-       Considering the subject of the paper, the introduction should have mentioned the main cultural characteristics of scholar environmental.

- Authors’ response: Thanks for your comment. Thus, we have include the following information into the introduction: “Thus, such practices become even more important and necessary in a school environ-ment where students spend around 6 hours physically inactive sitting on a desk; only having a 30-min break and the 5-min times between classes as free time to have active behaviours, in addition to the two hours per week of Physical Education classes”.

-       I suggest a figure with a schematic representation of a test day to understand correctly the work and procedures.

                - Authors’ response: Following your suggestion, we have included a figure with the scheme of the active break development.

Results.

-       Results: I was elaborated well. In this case, the manuscript providing immediate results inside the classroom with cognitive task, something different from what is found in the literature that normally use questionnaires. The results of ANOVA for RT and RPE were quite clarifying and robust.

                - Authors’ response: Thank you for your comment and recognition of our work.

-       Elucidate the table, add the confidence intervals 95% upper and lower and add note under the table to clarify the information.

                - Authors’ response: We appreciate your comment. However, we do not understand the exactly you want. Could you indicate us the table you refer and clarify to which parameter you want we add the 95% CI? Thank you in advance and sorry for any inconvenience.

Discussion

-       Discussions is appropriate, making reference to the results of other studies, but the theoretical and practical implications of the research are vaguely mentioned. It necessary improve it.

                - Authors’ response: thank you for your comment, we have added the information you recommended in the discussion section. If you feel that more information is needed, please let us know.

References

-       The references used in the present manuscript were sectioned correctly and help to further clarify the interest of the present research topic. Update and check the last two years

                - Authors’ response: citations of some recent reviews on the subject have been added, if you feel that something else should be included please let us know.

Other

-       Last, how would you start to be implementing the AB in classroom? Where should you start?

- Authors’ response: Thank you for your interest. Thus, we have added the following at the end of the conclusion section: “An adequate way to introduce and implement active breaks at school would be to carry out activities during the 5 minutes break between classes or by starting each class with a brief activation of the students in the classroom by performing activities to remind them of the contents taught in the previous class of that subject while they carry out motor activities. Another possibility would be to take advantage of the spaces in the school outside the classroom to design a route for students to go out for aerobic physical activity and practice basic motor skills for 8-10 minutes after long periods of class. Ideally, however, all teachers in the school should be trained to acquire basic knowledge about the concept and applications of active breaks so that they can easily implement it”.

-       Be consistent with the terms. i.e.: Active-breaks?

                - Authors’ response: Based on your comment and in order to make it easier to read we have replaced active breaks by AB.

Reviewer 2 Report

The authors propose a very interesting aspect of physical activity at school, I absolutely agree with the approach used, I think it is essential to have a pleasant aspect to make training motivating and productive, especially with children.

The progression also seems adequate and reasonable, I only have a few notes:

- The instrument used for the bioimpedance is not very reliable, even less for the age group considered, even if it is not the focus of the research.

- Is there a difference between males and females?

_ Was there a difference in school learning levels? (I would expect so ....)

Author Response

Dear Reviewer,

Thank you for your review of our manuscript. We have carefully considered your comments and believe that the quality of the paper has improved after incorporating your suggestions. Below are our responses to your suggestions:

The authors propose a very interesting aspect of physical activity at school, I absolutely agree with the approach used, I think it is essential to have a pleasant aspect to make training motivating and productive, especially with children.

The progression also seems adequate and reasonable, I only have a few notes:

- The instrument used for the bioimpedance is not very reliable, even less for the age group considered, even if it is not the focus of the research.

                - Author’s response: BIA offers an acceptable and reproducible alternative for assessing body composition in this population (Verney et al., 2016; Verney, Schwartz, Amiche, Pereira, & Thivel, 2015). However, it should be noted that the use of BIA in morbidly obese adolescents is unclear, as the correlation between BIA and DXA decreases as fat mass increases (Verney et al., 2016). Specifically, none of the participants in this study were morbidly obese, and therefore this does not affect our results. Also, as you mention it is not a variable that affects the results of the study, we used it for the characterisation of the sample.

If you consider that we should include some information in the masnuscript please let us know.

                Verney, J., Metz, L., Chaplais, E., Cardenoux, C., Pereira, B., & Thivel, D. (2016). Bioelectrical impedance is an accurate method to assess body composition in obese but not severely obese adolescents. Nutrition Research36(7), 663-670.             

                Verney, J., Schwartz, C., Amiche, S., Pereira, B., & Thivel, D. (2015). Comparisons of a multi-frequency bioelectrical impedance analysis to the dual-energy X-ray absorptiometry scan in healthy young adults depending on their physical activity level. Journal of human kinetics47(1), 73-80.

- Is there a difference between males and females?

                - Author’s response: No gender differences were detected in the PL scores or in any of its domains, neither for the experimental group (pre- and post-intervention) nor for the control group (initial and final assessment). This, together with the fact that other studies that have assessed PL at these ages have also reported no sex differences, led us to report the results without sex differences as it seems that the level of PL is not affected by sex at these ages. But if you think we should include it, please let us know.

                - Caldwell, H. A., Di Cristofaro, N. A., Cairney, J., Bray, S. R., MacDonald, M. J., & Timmons, B. W. (2020). Physical literacy, physical activity, and health indicators in school-age children. International journal of environmental research and public health, 17(15), 5367.

                - Mendoza Muñoz, M., López García, C., Franco García, JM, Calzada Rodríguez, JI, Denche Zamorano, Á. M., & Carlos Vivas, J. Valores de alfabetización física en niños con edades comprendidas entre 8 y 12 en Extremadura: Estudio piloto.

_ Was there a difference in school learning levels? (I would expect so ....)

                - Author’s response: There was an increase in the level of knowledge about physical activity in the experimental group. But, If you refer to general learning in other areas, this was not evaluated, but as we consider it would be very interesting to study, based on your comment, we have added the following line for the future. “Another line for the future that could be addressed is the effect of this programme on school learning in other areas, i.e. to see if there is an improvement in other areas such as language or mathematics when this type of activity is carried out during breaks”.

Round 2

Reviewer 1 Report

Thank you very much. The suggestion were aborded correctly and the paper would be accepted in the present version.